# Deep Spectral Clustering for Object Instance Segmentation

**Matt Barnes & Artur Dubrawski**
The Robotics Institute
Carnegie Mellon University
Pittsburgh, PA 15213, USA
{mbarnes1,awd}@cs.cmu.edu

## Abstract

We introduce a new graph Laplacian which is only available during training time and show how existing feedforward networks can learn to predict its eigenvectors. This is particularly useful for the problem of image segmentation, as it allows converting existing semantic segmentation networks (a basic multi-class classification problem) into instance segmentation networks (a more complex structured prediction problem). We show some convenient theoretical properties of this graph Laplacian, propose several rounding schemes (with error bounds) of the eigenvectors, and present preliminary results showing an insignificant change in error after converting from semantic to instance segmentation on the PASCAL VOC dataset.

## 1 Introduction

In this paper, we consider the problem of object instance segmentation in images. Unlike semantic image segmentation, which can be formulated as a multi-class classification problem and predicted using the single pass of a CNN with dense outputs (e.g. FCN (Long et al., 2015)), instance segmentation is a more complex structured prediction problem. Current state-of-the-art approaches to instance segmentation use a sequence of learned and hand-tuned components, including proposing multiple object regions, performing a series of binary classifications, refining the object region and fixing object overlap issues in post-processing (He et al., 2017). This is significantly more complex and ad-hoc than solutions to the semantic classification problem, which are able to learn end-to-end. Instead, we show how to retrain any existing semantic segmentation network with dense outputs for instance segmentation by using spectral techniques.

For much of the early 2000s, spectral clustering was considered the state-of-the-art approach to image segmentation (von Luxburg, 2007). The approach can be broken into three parts:

1. Choose a pixel similarity function $s : \mathcal{X} \times \mathcal{X} \to \mathbb{R}$, where $\mathcal{X}$ is the pixel space.
2. Form the graph Laplacian $L$ by performing pairwise comparisons between all pixels within some local neighborhood of size $m$ and compute its top eigenvectors.
3. Cluster the eigenvector rows with k-means.

A major factor in the performance of spectral clustering is the choice of similarity function $s$, which must capture whether two pixels belong to the same segment. A naïve approach to incorporating the representational power of deep learning into spectral clustering would be to model $k$ as a deep network, which would require a large number of $nm$ outputs.

Instead of choosing or learning a similarity function, we instead consider the oracle kernel $s(x_i, x_j) = 1$ if samples $x_i$ and $x_j$ belong to the same segment, else $s(x_i, x_j) = 0$. Note this is equivalent to having a set of training images with instance labels, and does not require semantic labels. We then train a deep network to predict the top eigenvectors of a particular graph Laplacian induced by $s$, effectively replacing steps 1 and 2 above with a deep network. The embedding provided by these eigenvectors naturally allows for variable number of instances, while the learner remains a simple, single feedforward network where we only change the final output layer. This is a powerful representational tool for instance segmentation.

The remainder of this paper is organized as follows. We begin in Section 2.1 with the introduction of a new graph Laplacian for the oracle kernel setting and prove some convenient properties. In Section 2.2, we explain how to train a learner to predict the eigenvectors by using the spectral loss. Then in Section 2.3 we discuss rounding algorithms and theoretical guarantees (i.e. a new step 3, above). Lastly, we demonstrate that deep spectral clustering is able to retrain an FCN-8 semantic segmentation network for instance segmentation with insignificant change in error on the PASCAL VOC dataset.

## 2 DEEP SPECTRAL CLUSTERING

Given an image $X = (x_1, \ldots, x_n)$ with $n$ pixels, our objective is to partition the pixels such that pixels in the same partition belong to the same object (potentially including a "background" object). Our approach is class-agnostic (i.e. does not require semantic labels), and has the potential to generalize to new classes.

### 2.1 GRAPH LAPLACIAN

Graph Laplacian matrices are the key to spectral graph theory, and we leverage their properties to perform image segmentation. Several variations have been proposed for spectral clustering, and here we propose a new graph Laplacian for the oracle kernel setting.

**Theorem 2.1.** *Consider the graph Laplacian $L = AD$, where $A$ is the adjacency matrix formed by the oracle kernel $s$ and $D$ is the diagonal degree matrix. Then the following properties hold:*

1. *$L$ is symmetric and positive semi-definite.*

2. *$L$ has $k$ positive, real valued eigenvalues $\lambda_1, \ldots, \lambda_k$ which correspond to the squared cardinality of the $k$ clusters. The other eigenvalues are 0.*

3. *$L$ has the partial spectral decomposition $L = V_o \Lambda V_o^\mathsf{T}$, where the columns of $V_o \in \mathbb{R}^{n \times k}$ are a set of orthonormal eigenvectors of $L$ and $\Lambda = \mathrm{diag}(\lambda_1, \ldots, \lambda_k)$ is a diagonal matrix of the corresponding non-zero eigenvalues.*

4. *Let $V = V_o \Lambda^{1/4}$ be orthogonal eigenvectors normalized according to their eigenvalues and $v_i$ denote the $i$'th row of $V$. Then $v_i^\mathsf{T} v_j = 1$ if $x_i$ and $x_j$ belong to the same cluster, else 0.*

In particular, we will leverage properties 3 and 4 for training the network and rounding the results.

### 2.2 OBJECTIVE FUNCTION

Let $\|\cdot\|$ be any unitarily invariant norm and $f : \mathbb{R}^n \to \mathbb{R}^{n \times d}$ be our learner, which maps every pixel into a $d$-dimensional space. In the case of semantic segmentation, $d$ is the number of object classes. By the Eckart-Young-Mirksy theorem, the optimal solution to

$$\underset{f}{\mathrm{minimize}} \quad \ell(f(X), A) = \|f(X)f(X)^\mathsf{T} - A\|$$

is $f^*(X) = V$, the normalized eigenvectors of $L$ (Eckart & Young, 1936; Mirsky, 1960). Thus, we train $f$ using $\ell$, which we term the *spectral loss*. In practice, we compute an unbiased estimator of the spectral loss by randomly sampling pixels, otherwise the matrices would have $n^2$ entries. Note that the solution is invariant under rotation, which does not restrict the learner to an arbitrarily chosen hot-one encoding.

### 2.3 ROUNDING

The top-$d$ predicted eigenvectors $f(X) = \hat{V}$ of an image provide a pixel representation such that the dot product between $\hat{v}_i$ and $\hat{v}_j$ measures the similarity between pixels $x_i$ and $x_j$. Thus, in the final step of deep spectral clustering, we cluster in the inner-product space of $\hat{v}_1, \ldots, \hat{v}_n$. This is analogous to the Euclidean space clustering step of traditional spectral clustering algorithms. For now, we choose to run KwikCluster, as it is linear-time in $n$ and provides straightforward theoretical guarantees (Ailon et al., 2008). We will explore other rounding schemes prior to the workshop.

| | Semantic | | | Instance | | |
|---|---|---|---|---|---|---|
| | Accuracy | Precision | Recall | Accuracy | Precision | Recall |
| FCN-8 | 0.859 | 0.864 | 0.915 | - | - | - |
| Deep Spectral | - | - | - | 0.852 | 0.848 | 0.919 |

Table 1: Deep spectral clustering, using the same FCN-8 network architecture, achieves almost identical performance on the more difficult instance segmentation task. We measure performance using pairwise accuracy, precision and recall, which do not require computing or thresholding IoU scores and only depend on the pixel partition.

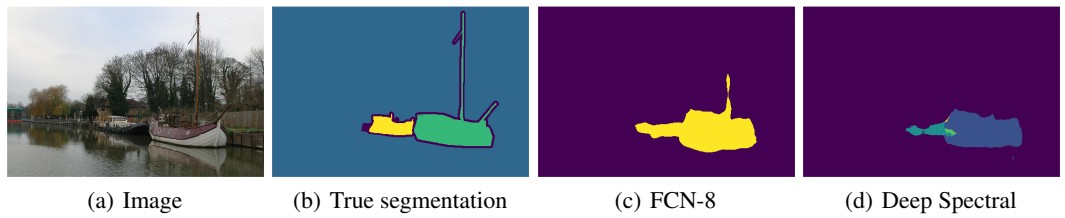

     (a) Image        (b) True segmentation       (c) FCN-8       (d) Deep Spectral

Figure 1: Segmentation results on a validation image.

**Theorem 2.2.** *Let $\hat{V} = f(X)$ be the predicted eigenvectors, $V_r$ denote the KwikCluster rounded eigenvectors (i.e. a hot-one encoding of every pixel) and $V_r^*$ the optimal rounding algorithm. Then the final loss is bounded by*

$$\ell(V_r V_r^\mathsf{T}, A) \leq \ell(\hat{V}\hat{V}^\mathsf{T}, A) + 5\ell(\hat{V}\hat{V}^\mathsf{T}, V_r^* V_r^{*T})$$
$$\leq 6\ell(\hat{V}\hat{V}^\mathsf{T}, A)$$

where $\ell(\hat{V}\hat{V}^\mathsf{T}, A)$ is the validation loss of the network. It is theoretically possible, though practically difficult, to decrease the constant factors 5 and 6 to 2.5 and 3.5, respectively, by solving a large linear programming variation of KwikCluster (Ailon et al., 2008).

## 3   Experimental Analysis

We use an FCN-8 network for $f$ and replace the softmax layer with a unit normalization layer, such that $\|\hat{v}_i\| = 1$ (Long et al., 2015). For fair comparison, we choose $d = 21$ such that our network has exactly the same number of parameters as the original FCN-8 architecture. We use the Frobenius norm $\|\cdot\|_F^2$ as our unitarily invariant matrix norm and increase the learning rate to $1e-5$ to accommodate changing the loss function from cross-entropy to spectral.

Our results, in Table 1 and Fig. 1, demonstrate that deep spectral clustering is able to retrain a semantic segmentation network for instance segmentation with an insignificant change in error.

## 4   Conclusions

We introduced the spectral loss function, which enables training existing dense output CNN's for the more complex variable output problem. Preliminary results are promising and we intend to additionally evaluate on more complex instance segmentation datasets (e.g. CityScapes and MS-COCO) with more baselines and metrics. The KwikCluster rounding scheme is fast and provides clear theoretical guarantees, though we believe it is possible to do much better in practice by solving for an locally optimal orthogonal projection.

### Acknowledgments

This work has been partially supported by DARPA under awards FA8750-14-2-0244 and FA8750-17-2-0212.

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
