# OpenReview forum: "Deep Spectral Clustering for Object Instance Segmentation"
_ICLR.cc/2018/Workshop — Reject_

### Official Review · AnonReviewer2 · 2018-03-08
**Deep Spectral Clustering for Object Instance Segmentation**

**Rating:** 6
**Confidence:** 3

**Review:**

The authors present a novel method for deep spectral clustering. It makes use of a kernel oracle, combined with training a deep network to predict the top eigenvectors of a graph Laplacian. The objective function for training is done in view of the Eckart-Young theorem. Overall, it provides an original new look at the problem with a sound methodology, supported also by two theorems.

Additional comments and suggestions:
- it would be interesting if the authors could give more information on the model selection aspects, e.g. the choice of d.
- more extensive comparisons in Table 1 would be good to see. From the current Fig 1 and Table 1 it is difficult to assess the relevance and significance of the method.
- the reference list is quite minimal. Other methods for deep spectral clustering have been proposed in literature. The method also relates to kernel methods. At this point work has been done already e.g. on kernel spectral clustering.

see e.g.

Spectral Networks and Locally Connected Networks on Graphs
Joan Bruna, Wojciech Zaremba, Arthur Szlam, Yann LeCun, arXiv:1312.6203

Mall R., Langone R., Suykens J.A.K., Multilevel Hierarchical Kernel Spectral Clustering for Real-Life Large Scale Complex Networks, PLOS One, e99966, vol.9, no. 6, pp. 1-18, 2014.

Deep Spectral Clustering Learning
Marc T. Law, Raquel Urtasun, Richard S. Zemel, ICML 2017

Alzate C., Suykens J.A.K., Hierarchical Kernel Spectral Clustering, Neural Networks, 35: 21-30, 2012

---

### Official Review · AnonReviewer3 · 2018-03-11
**review: insuffienctly developed for a workshop paper**

**Rating:** 2
**Confidence:** 5

**Review:**

This paper proposes to build a neural network for instance segmentation by training it to predict eigenvectors of the graph Laplacian induced by the ground-truth instance grouping. The CNN is thus trained to emulate a traditional spectral clustering approach to segmentation. Preliminary results are shown using a modification of the FCN-8 architecture.

While the proposed research direction has promise, the current contribution is not sufficiently developed to a state appropriate for even a workshop. Moreover, this paper is missing citation to a large array of related work, including that on both fundamental concepts, as well as recent publications which should serve as baselines for comparison.

In particular, the paper fails to:

(1) Cite and discuss prior work on spectral embedding approaches to image segmentation. For example:

Shi and Malik. Normalized Cuts and Image Segmentation. PAMI, 2000.

Arbelaez, Maire, Fowlkes, and Malik. Contour Detection and Hierarchical Image Segmentation. PAMI, 2011.

Bertasius, Shi, and Torresani. High-for-Low, Low-for-High: Efficient Boundary Detection from Deep Object Features and its Applications to High-Level Vision. ICCV, 2015.

Maire, Narihira, Yu. Affinity CNN: Learning Pixel-Centric Pairwise Relations for Figure/Ground Embedding. CVPR, 2016

(2) Cite and discuss prior work on training (e.g. via backprop) through spectral embedding:

Cour, Gogin, Shi. Learning Spectral Graph Segmentation. AISTATS, 2005.

Ionescu, Vantzos, Sminchisescu. Matrix Backpropagation for Deep Networks With Structured Layers. ICCV, 2015.

(3) Cite recent work on other embedding approaches to object instance segmentation. For example:

Bai and Urtasun. Deep Watershed Transform for Instance Segmentation. CVPR, 2017.

Newell, Huang, Deng. Associative Embedding: End-to-End Learning for Joint Detection and Grouping. NIPS, 2017.

Kong and Fowlkes. Recurrent Pixel Embedding for Instance Grouping. arXiv:1712.08273

(4) Compare to at least one of the recent publications (such as those listed above) that combines neural networks with embedding methods for instance segmentation. This prior work is an appropriate and necessary experimental baseline against which to view the proposed method. FCN-8 is not a sufficient quantitative baseline in light of these more directly related publications.

In light of the other embedding methods, it is also not clear why the graph Laplacian is necessary as a training target. Why not simply impose a softer requirement that pixels that should be grouped together map to similar locations in the embedding space, but map to distant locations otherwise? What advantage does emulating spectral clustering offer?

(5) Provide a more complete visualization of the results of the proposed method, and a visual comparison to a proper baseline method as discussed in point (4). Showing output on only a single example image is not informative.

---

### Official Review · AnonReviewer1 · 2018-03-11
**A fully supervised approach to performing instance segmentation in images**

**Rating:** 6
**Confidence:** 5

**Review:**

Summary:
Have proposed a fully supervised approach to performing instance segmentation in images. The paper claims that instance segmentation is more challenging than semantic segmentation.
The idea is to approximate the eigenvectors of the “ideal” Laplacian matrix (where k(x_i, x_j) = 1, if pixels x_i and x_j belong to the same object and 0 otherwise using a neural network by using the rank k reconstruction loss (k is the number of components)
The paper claims that any method that performs semantic segmentation by solving a multiclass classification problem, can be modified in the last layer to generate eigenvectors instead. The authors use FCN-8 (a network architecture used for semantic segmentation) network architecture.
The performance of both original FCN-8 and the proposed method is same on PASCAL VOC dataset, however since the authors claim the instance segmentation is harder, this is a good thing.

Issues:
More experiments are needed to justify the validity of the proposed method.
How will the number of objects be chosen in a new test image?

Other Interesting Points:
The authors use KwikCluster algorithm (and not k-means) on the approximation to the similarity matrix obtained from the algorithm to finally cluster pixels. I think there are two advantages of this:
Individual eigenvectors need not be perfect. In fact, they need not be eigenvectors at all if they are sufficiently good at reconstructing the similarity matrix S. They are not using eigen embeddings for clustering
KwikCluster runs in linear time as opposed to k-means and hence it is computationally better.
One downside is that KwikCluster is a randomized approximation algorithm that may perform very poorly due to randomness.

Overall Opinion: Marginally above acceptance threshold. Good idea but not enough supporting experiments. Needs fully supervised pixel by pixel instance labels during training.

---

### Decision · Program_Chairs · 2018-03-20
**ICLR 2018 Workshop Acceptance Decision**

**Decision:**

Reject

**Comment:**

Based on the reviews, this paper has not been accepted for presentation at the ICLR workshop. However, the conversation and updates can continue to appear here on OpenReview.